# Probing the gating mechanism of the mechanosensitive channel Piezo1 with the small molecule Yoda1

Jerome J. Lacroix[1], Wesley M. Botello-Smith[2] & Yun Luo [2]

Piezo proteins are transmembrane ion channels which transduce many forms of mechanical stimuli into electrochemical signals. Their pore, formed by the assembly of three identical subunits, opens by an unknown mechanism. Here, to probe this mechanism, we investigate the interaction of Piezo1 with the small molecule agonist Yoda1. By engineering chimeras between mouse Piezo1 and its Yoda1-insensitive paralog Piezo2, we first identify a minimal protein region required for Yoda1 sensitivity. We next study the effect of Yoda1 on hetero-trimeric Piezo1 channels harboring wild type subunits and Yoda1-insensitive mutant subunits. Using calcium imaging and patch-clamp electrophysiology, we show that hybrid channels harboring as few as one Yoda1-sensitive subunit exhibit Yoda1 sensitivity undistinguishable from homotrimeric wild type channels. Our results show that the Piezo1 pore remains fully open if only one subunit remains activated. This study sheds light on the gating and pharmacological mechanisms of a member of the Piezo channel family.

[1] Graduate College of Biomedical Sciences, Western University of Health Sciences, 309 E. Second St, Pomona, CA 91766, USA. [2] College of Pharmacy, Western University of Health Sciences, 309 E. Second St, Pomona, CA 91766, USA. Correspondence and requests for materials should be addressed to J.J.L. (email: jlacroix@westernu.edu)

The transduction of mechanical stimuli into biological signals, or mechanotransduction, is a ubiquitous phenomenon observed in every known biological kingdom. Mechanotransduction regulates many important homeostatic and sensory functions from the cellular to the organ level. In vertebrate organisms, the recently identified mechanosensitive ion channels Piezo1 and Piezo2 participate in a bewildering number of mechanotransduction processes[1]. These include touch sensation[2–4], proprioception[5], hearing[6], vascular[7,8] and brain development[9], blood flow sensing[10], osmotic homeostasis[11], and epithelial cell number regulation[12,13]. In light of their physiological importance, both gain-of-function and loss-of-function mutations of Piezo channels have been associated with pathological conditions such as xerocytosis[11,14–16], arthrogryposis[17–23], and lymphedema[24]. Recent studies suggest Piezo channels may also have important roles in other conditions such as sleep apnea[25] and hyperalgesia[26–29].

Piezo proteins have no homology with any known protein family[2] and possess a homotrimeric structure that resembles a propeller with three peripheral blades and a central cation-selective pore[30–33] (Fig. 1a). Structure–function studies have shown that Piezo1 can be directly activated by membrane stretch in absence of other cellular components[34], suggesting that Piezo1 directly senses forces transmitted from lipids such as tension or curvature[1,30]. Although the identification of specific protein regions involved in mechanosensing is emerging[33], the mechanism by which mechanical forces are detected by Piezo channels and transmitted to the pore is currently unclear.

Interestingly, Piezo1 can also be activated by a synthetic small molecule called Yoda1[35]. Yoda1 stabilizes the open conformation of the channel, reducing the mechanical threshold for activation. This effect allows Yoda1 to partially activate Piezo1 channels in absence of mechanical stimulation. Interestingly, Piezo2 is not modulated by Yoda1, despite the high sequence similarity between these two homologs.

Here, we use Yoda1 as a probe to investigate the gating mechanism of Piezo1 channels. On the basis of the strict selectivity of Yoda1 between mammalian Piezo1 and Piezo2 channels, we used a chimeric approach to identify a minimal region required to transduce the effect of the agonist and engineered a Yoda1-insensitive Piezo1 chimera. We then characterize hybrid channels obtained by mixing Yoda1-sensitive wild type (WT) subunits with Yoda1-insensitive chimeric subunits. Interestingly, our analysis indicates that the presence of a single agonist-sensitive subunit per channel is sufficient to mediate Yoda1-induced pore opening.

## Results

### Design of a Yoda1-insensitive Piezo1 chimera.
As Yoda1 activates Piezo1 but not the Piezo2 homolog, some Piezo1 residues involved in mediating the effects of Yoda1 must be absent in Piezo2. It should therefore be possible to engineer a Yoda1-insensitive Piezo1-Piezo2 chimera. Since chimeras between mouse Piezo1 (mPZ1) and the evolutionary-distant drosophila channel dPiezo produce functional channels[36], we can expect mouse Piezo1-Piezo2 chimeras to yield functional channels as well. Hence, we engineered three C-terminal chimeras between mPZ1 and mouse Piezo2 (mPZ2). These chimeras were named Ch1961, Ch2063 and Ch2456 according to the position of the most C-terminal residue from mPZ1 (Fig. 1b and Supplementary Figure 1). These chimeras were cloned into a pCDNA3 plasmid and expressed into Human Embryonic Kidney (HEK) cells 293T. Since Piezo channels activation produces intracellular $Ca^{2+}$ influxes, we tested the ability of Yoda1 to activate the chimeras using time-lapse calcium imaging. To this aim, cells were pre-incubated with the calcium-sensitive cell-permanent dye Fluo8-AM and imaged with an inverted fluorescence microscope during acute Yoda1 perfusion. In control experiments, the intracellular calcium concentration increases when mPZ1-expressing cells were exposed to Yoda1 but not when either Yoda1 was absent or when cells were transfected with a mock plasmid (Fig. 1c). Exposure of mPZ1, Ch2063, and Ch2456 to the agonist yielded relative fluorescence changes (($F_{t=1min} - F_{t=0})/F_{t=0}$ or $\Delta F/F_0$) that increased gradually with increasing Yoda1 concentrations, indicating that Ch2063 and Ch2456 remain Yoda1-sensitive, similar to WT mPZ1 channels (Fig. 1d). Thus, the mPZ1 region from residues 2063 to the C-terminal end does not appear to contain Piezo1-specific molecular determinants required to mediate the agonist effect of Yoda1.

In contrast, Ch1961 did not respond to any tested Yoda1 concentration and even displayed negative $\Delta F/F_0$ values, most likely due to bleaching of the Fluo8 chromophore as evidenced from Fig. 1c (blue and magenta plots). A lack of Yoda1-induced calcium response could be caused by a loss-of-function of the chimeric channel rather than a loss of sensitivity to the agonist. To test whether this was the case, the chimeras were functionally tested based on their ability to evoke calcium signals in response to membrane stretch induced by reducing extracellular osmolarity by ~250 mOsmol $L^{-1}$ (see Methods). The $\Delta F/F_0$ values produced using this assay were significantly larger in cells expressing mPZ1 or any chimera that in cells transfected with an empty pCDNA3 vector (Fig. 1e). This indicates that the three chimeras form functional, stretch-activated calcium channels.

These results suggest that the region 1961–2063 contains important molecular determinants to mediate the effects of the agonist. To verify this hypothesis, we engineered the internal chimera Ch1961–2063, which substitutes the region 1961–2063 of mPZ1 by its homologous counterpart from mPZ2. We also created additional chimeras within the 1961–2063 region by dividing it into three sub-domains (sub-domain 1: 1961–2004, sub-domain 2: 2005–2034 and sub-domain 3: 2035–2063). Six chimeras were made to create every sub-domain combinations (Fig. 1f and Supplementary Figure 1). For simplicity, these chimeras were identified by the name of the sub-domain(s) being substituted (e.g., Ch1+3 replaces both mPZ1 sub-domains 1961–2004 and 2035–2063 by their homolog counterparts from mPZ2). To reduce background fluorescence, calcium signals were further detected with the genetically encoded calcium indicator GCaMP6m (GC6) co-transcriptionally expressed with each tested Piezo channels and chimeras. Time-lapse imaging experiments show that, similar to Fluo8, GC6 fluorescence increases upon Yoda1 application in the presence, but not in absence, of mPZ1 (Supplementary Movies 1–2).

The internal chimera Ch1961–2063 displayed negative or nearly null $\Delta F/F_0$ values upon any tested agonist concentrations, showing that the 103 amino acids in the 1961–2063 region contain necessary determinants to mediate chemical activation of Piezo1 (Fig. 1g). In contrast, the sub-domain chimeras produce heterogeneous Yoda1 dose–responses. At the maximum tested Yoda1 concentration of 100 μM, the calculated $\Delta F/F_0$ values for the chimeras were $2.66 \pm 0.22$ for Ch1 ($n = 4$), $2.09 \pm 0.17$ for Ch2 ($n = 4$), $1.60 \pm 0.13$ for Ch3 ($n = 4$), $1.04 \pm 0.1$ for Ch1 + 2 ($n = 4$), $0.6 \pm 0.08$ for Ch2 + 3 ($n = 4$) and $1.59 \pm 0.14$ for Ch1 + 3 ($n = 4$). These values were higher than the $\Delta F/F_0$ values obtained from Ch1961–2063 but lower than for WT mPZ1 (Fig. 1g). In addition, the lack of Yoda1 sensitivity in Ch1961–2063 was not due to a loss of function, as evidenced by the ability of Ch1961–2063 to elicit $Ca^{2+}$ influxes upon acute hypotonic shocks (Fig. 1h). Interestingly, Ch3 produced the strongest loss of Yoda1 sensitivity by individual sub-domain replacement, while Ch2+3 produced the largest loss of Yoda1 sensitivity of any tested sub-domain chimera. These results suggest that the C-terminal

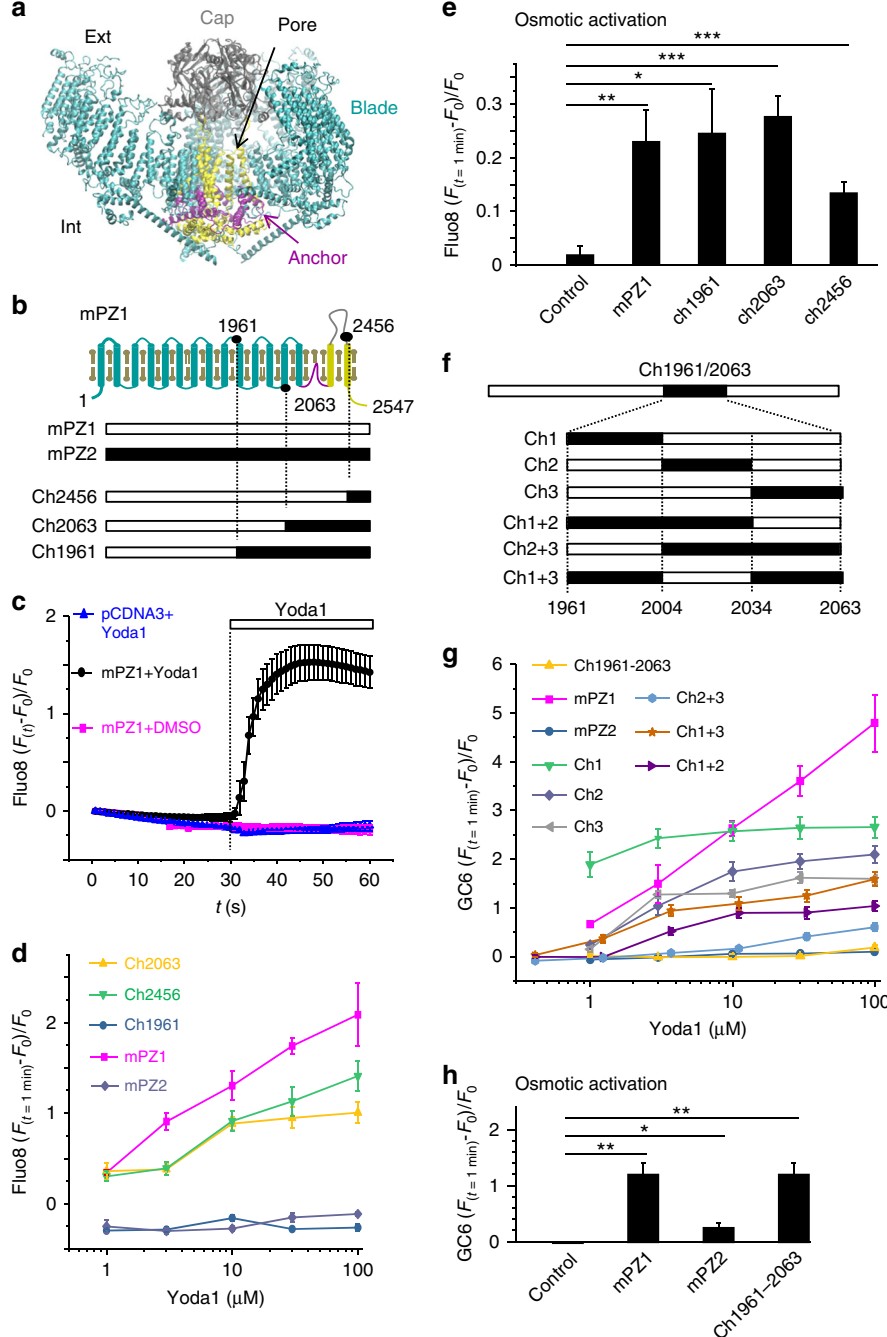

**Fig. 1** Design of a Yoda1-insensitive Piezo1 chimera. **a** mPZ1 cryo-EM structure (PDBID:6B3R) created using Molecular Visual Dynamics (VMD) with structural features highlighted with differential coloring. **b** Generation of C-terminal Piezo chimeras with respect to their approximate position with a simplistic mPZ1 topology (coloring identical to **a**, residues position not to scale). Residue numbers are from mPZ1. **c** Relative $Ca^{2+}$-sensitive fluorescence time course in HEK 293T cells transfected with WT mPZ1 (black and magenta traces) or the empty vector pCDNA3 (blue trace). Cells were pre-loaded with Fluo8-AM and incubated with 30 μM Yoda1 (blue and black traces) or a control solution (magenta trace) at $t = 30$ s. **d** Relative fluorescence changes from HEK 293T cells expressing the indicated constructs plotted against Yoda1 concentration. **e** Relative fluorescence changes from HEK 293T cells transfected with the indicated construct or with the empty vector pCDNA3 following an acute hypotonic shock. **f** Generation of internal Piezo chimeras. Residue numbers are from mPZ1. **g** Relative changes of $Ca^{2+}$-sensitive fluorescence measured using GCaMP6m (GC6) for the indicated Piezo channels and chimeric variants and plotted as a function of Yoda1 concentration. **h** Relative fluorescence changes from HEK 293T cells expressing the indicated construct or the empty vector pCDNA3 following an acute hypotonic shock. In **d** and **g**, the experiment was done in duplicate and repeated four times. In **e** and **h**, the experiment was done in duplicate and repeated five times and the fluorescence signal was compared to the control using a two-tailed unpaired Student's $t$-test. The asterisks indicate standard $p$-value range: *: $0.01 < p < 0.05$; **: $0.001 < p < 0.01$ and ***: $p < 0.001$. In **c–e** and **g**, **h**, error bars = s.e.m

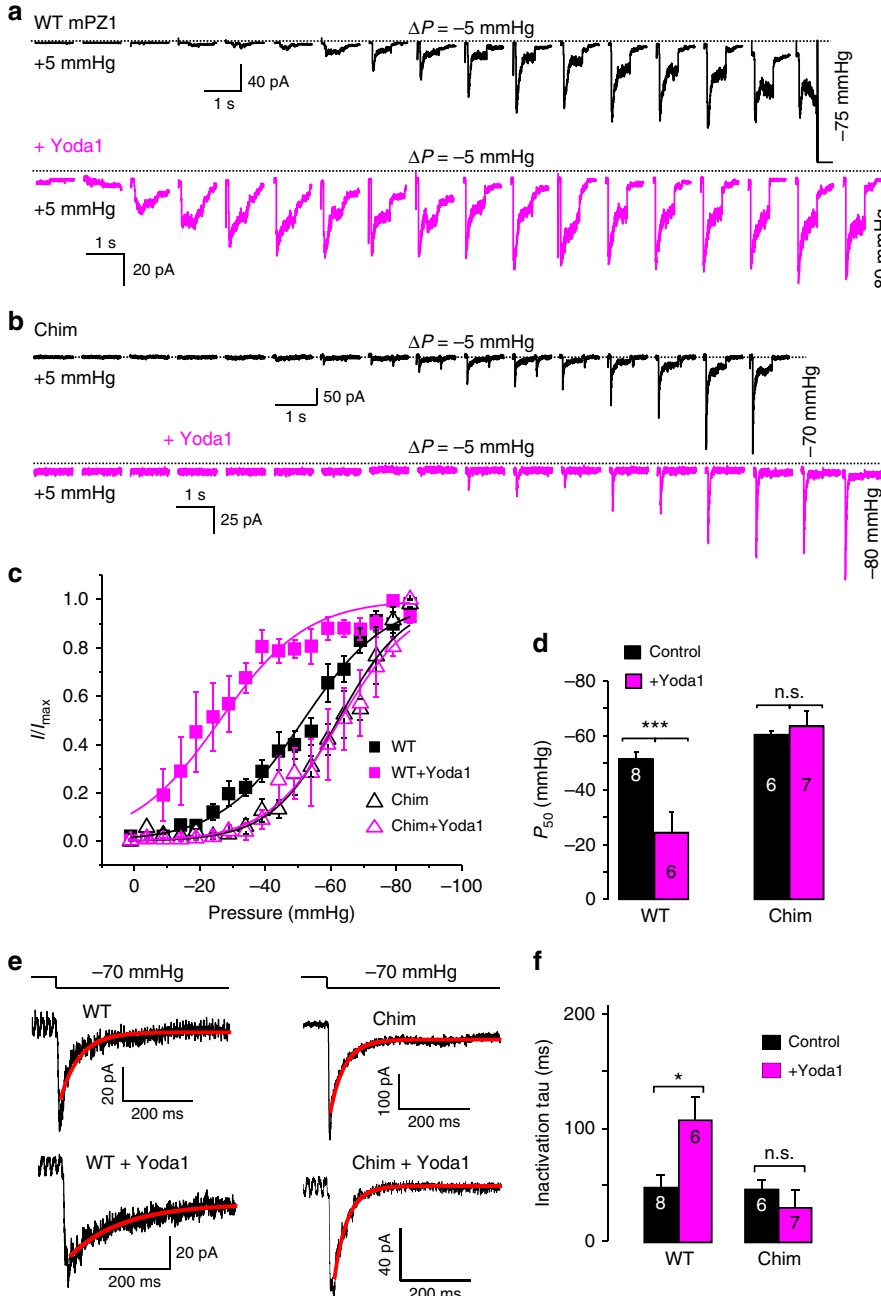

**Fig. 2** The Piezo chimera 1961–2063 is not modulated by Yoda1. **a, b** Representative family of pressure-evoked current traces recorded in the cell-attached configuration for WT mPZ1 (**a**) or the chimera 1961–2063 (Chim) (**b**) in the presence (magenta traces) or absence (black traces) of 30 μM Yoda1 in the patch pipette. Dotted lines indicate zero current level. **c** Normalized peak current plotted as a function of the pressure pulse for WT mPZ1 (full squares) and Chim (open triangles) in the presence (magenta squares and triangles) or absence (black squares and triangles) of 30 μM Yoda1. The traces correspond to a fit with Eq. 1. The plots correspond to the average of at least 5 independent recordings. **d** Histogram showing average $P_{50}$ values for WT and Chim in presence (magenta bars) or absence (black bars) of 30 μM Yoda1. **e** Example of curve fitting of decaying currents for the indicated constructs and pressure pulse using a monoexponential decay function. **f** Histogram showing the fitted inactivation time constant (tau) for WT and Chim in presence (magenta) or absence (black) of 30 μM Yoda1. For all electrophysiology data, $V = -80$ mV. The number in each histogram indicates the number of independent traces analyzed. For each histogram, a Student $t$-test was used to test if the presence of Yoda1 significantly changes $P_{50}$ or tau: n.s. non-significant; ***: $p <$ 0.001***: **c**, **d** and **f** error bars = s.e.m.

part of the 1961–2063 region has an important role in imparting Yoda1 sensitivity to the Piezo1 channel.

**Ch1961–2063 is not activated by Yoda1.** Unfortunately, it is inherently difficult to quantitatively correlate differences in Δ$F$/$F_0$ values to differences in Yoda1 sensitivity. Indeed, the calcium-dependent Δ$F$/$F_0$ values obtained from our assay are

not only a function of the interaction between the agonist and the channel protein. They also depend of the total number of channels expressed at the cell membrane, the biophysical properties of the chimeras and the intrinsic calcium-handling properties of each cell being measured. In particular, the detection of Piezo1 activation by Yoda1 in a calcium imaging experiment is possible because Yoda1 decreases the channel

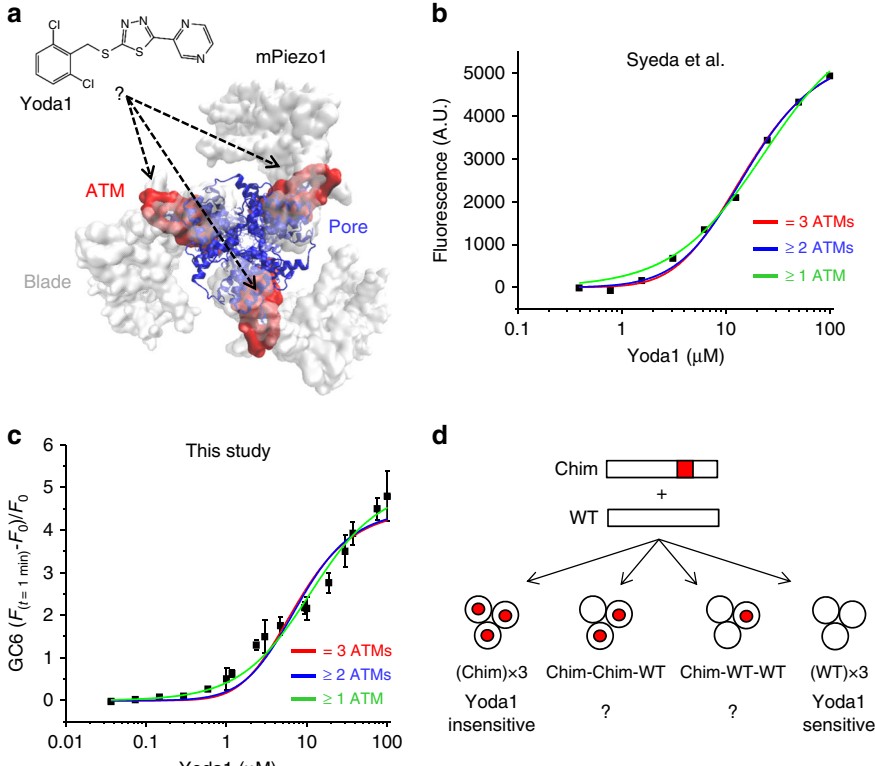

**Fig. 3** Experimental strategy to determine the gating mechanism of Piezo1. **a** mPiezo1 cryo-EM structure (PDBID:6B3R, image created using VMD) viewed from the intracellular side and depicting the blade (white surface), the pore (blue cartoons) and the ATM (red surface) in each subunit. The interaction between the ATM and Yoda1 (represented using ChemDraw) is unknown. Chemical gating of Piezo1 may require the activation of at least one, at least two, or all three ATM. **b**, **c** Curve fitting of Yoda1 dose–responses obtained from Syeda et al. (**b**) or from us (**c**) using models describing the interaction of the agonist with one or more (green), two or more (blue) or three (red) ATM. These models correspond to fits with Eqs. 2, 3, and 4, respectively. All three models were able to fit well the data (Supplementary Table 2). **d** A minimum ATM stoichiometry for chemical gating of Piezo1 could be determined by characterizing channels harboring Yoda1-sensitive (WT) and Yoda1-insensitive (Chim) subunits. In panels c, the experiment was done in duplicate and repeated five times. error bars = s.e.m.

mechanical threshold in such a way that it becomes able to sojourn into its open state for a significant fraction of the time in absence of external mechanical stimuli[35]. Hence, if the mechanical sensitivity of a Yoda1-sensitive chimera is intrinsically higher than WT mPZ1, the effect of Yoda1 on this chimera may not be sufficient to increase open probability in absence of external forces. Such scenario could exist for Ch1961–2063 (Chim) which exhibits an apparent loss of Yoda1 sensitivity. Hence, to examine more quantitatively the effect of Yoda1 on Chim, we performed on-cell pressure-clamp electrophysiology recordings (Fig. 2a, c). In these experiments, channel opening is evoked using brief negative pressure pulses applied to the backside of a patch pipette (2–3 μm tip diameter) while maintaining a patch potential of −80 mV relative to the inside of the cell. To compare the results with other groups, we used experimental settings identical to those used by Syeda et al.[35] For each recording, the normalized peak current $I/I_{max}$ values were plotted against the applied pulse pressure and the plot was fitted with a standard two-state Boltzmann equation:

$$\frac{I}{I_{max}} = \frac{1}{1 + e^{[(P-P_{50})/k]}} \quad (1)$$

with $P$ the pulse pressure in mmHg, $P_{50}$ the pressure corresponding to $I = I_{max}/2$ and the slope factor $k$, a fitted constant expressed in mmHg that reflects the electromechanical coupling of the protein. The fitted parameters values were obtained by fitting the average $I/I_{max}$ vs. pressure plot for WT and Chim

in presence and absence of Yoda1 (Supplementary Table 1). We also calculated the mean $P_{50}$ values by averaging the $P_{50}$ obtained from fitting individual $I/I_{max}$ vs. pressure plots (Fig. 2d). In this case, the mean $P_{50}$ values were −51 ± 3 mmHg for WT in absence of Yoda1 ($n = 8$) and −25 ± 8 mmHg with 30 μM Yoda1 ($n = 6$). For Chim, the mean $P_{50}$ values were −59 ± 1 mmHg without Yoda1 ($n = 6$) and −63 ± 6 mmHg with 30 μM Yoda1 ($n = 7$). Hence, while $P_{50}$ was significantly reduced by Yoda1 in mPZ1 (t-test p-value = 0.003), it did not significantly change in Chim (t-test p-value = 0.48). This shows Chim is truly Yoda1-insensitive as mPZ2[35]. In absence of the agonist, the chimera produces a small negative-shift of pressure sensitivity (i.e., $P_{50}^{Chim} = -59 \pm 1$ mmHg vs. $P_{50}^{WT} = -51 \pm 3$ mmHg). Such a moderate alteration of pressure sensitivity is not too surprising since the chimera is located near the anchor region, a domain postulated to couple the mechanosensory machinery of Piezo channels to their central pore.

Besides reducing the pressure sensitivity of Piezo1, Yoda1 also slows down channel inactivation[35]. We thus compared the time course of ionic currents evoked with a −70 mmHg pressure pulse in cells transfected with WT mPZ1 and Chim in presence or absence of Yoda1 (Fig. 2e, f). Our data show that the rate of inactivation of WT mPZ1 channels in absence of Yoda1 (48 ± 11 ms, $n = 4$) increases significantly upon Yoda1 exposure (107 ± 20 ms, $n = 4$). In contrast, the inactivation time course for Chim (46 ± 8 ms, $n = 4$) did not significantly change in the presence of Yoda1 (30 ± 16 ms, $n = 4$).

**Testing Piezo1 gating models**. The Piezo1 region 1961–2063, named Agonist Transduction Motif (ATM) is located at the interface between the blade and pore domains in each subunit (Fig. 3a). This indicates that the Piezo1 channel possesses three potential Yoda1 interacting regions, one in each subunit. Determining the minimal number of Yoda1-activated subunits needed for chemical pore opening would give us precious mechanistic insights into the gating mechanism of Piezo1. To answer this question, we decided to perform curve fitting of Yoda1 dose–responses using the classical equations that describe the interaction of a ligand with multiple protein binding sites[37] (Supplementary Equations 8, 9 and 10). Although curve fitting with these equations is originally intended to study ligand-binding sites, we used them instead to determine the number of Yoda1-activated subunits needed for channel activation. If the interaction of Yoda1 with one, two or three ATM (≥1 ATM) is sufficient for channel activation, the Yoda1 dose–response can be described with the following equation:

$$\frac{\Delta F}{F_0} = B_{max} \times \frac{3[L]}{K_d + [L]} \qquad ;(\geq 1\,\text{ATM}) \qquad (2)$$

with $\Delta F/F_0$ the relative fluorescence change, $[L]$ the Yoda1 concentration, $K_d$ the apparent dissociation constant for the agonist and $B_{max}$ a constant corresponding to the maximal fluorescence signal obtainable. If two or more activated ATM are needed (≥2 ATM), the fluorescence signal follows:

$$\frac{\Delta F}{F_0} = B_{max} \times \frac{\frac{6[L]^2}{K_d^2} + \frac{3[L]^3}{K_d^3}}{1 + \frac{3[L]}{K_d} + \frac{3[L]^2}{K_d^2} + \frac{[L]^3}{K_d^3}} \qquad ;(\geq 2\,\text{ATM}) \qquad (3)$$

If three activated ATM are required (≥3 ATM), the dose–response will follow:

$$\frac{\Delta F}{F_0} = B_{max} \times \frac{\frac{3[L]^3}{K_d^3}}{1 + \frac{3[L]}{K_d} + \frac{3[L]^2}{K_d^2} + \frac{[L]^3}{K_d^3}} \qquad ;(\geq 3\,\text{ATM}) \qquad (4)$$

We tested these three models by fitting the Yoda1 dose–responses from Syeda et al. and from our lab with Eqs. 2, 3 and 4. Surprisingly, all three models were able to fit the plots well and with similar coefficients of determinations (Fig. 3b, c and Supplementary Table 2). Hence, the point-by-point resolution of these experimental plots does not allow us to clearly eliminate or validate either of these gating models.

**Characterization of hybrid channels**. Since curve fitting of dose–response curves from WT channels did not allow us to answer our question, we next decided to study hybrid channels containing both Yoda1-sensitive (WT) and Yoda1-insensitive (Chim) subunits (Fig. 3d). The best way to construct such channels would be to concatenate WT and Chim subunits in a desired stoichiometry. However, the Piezo1 subunits are extremely large (2547 amino acids), making it very difficult to link three subunits into the same genetic vector. In addition, the distance between the N and C termini in two adjacent subunits may be long, precluding concatenation without a long unnatural linker. To circumvent this issue, we instead postulated that a population of hybrid channels containing predictable subunit stoichiometry could be obtained by transfecting HEK 293T cells with ratiometric mixtures of plasmid encoding WT and Chim subunits. To verify that this was the case, we inserted an enhanced

Green Fluorescent Protein (eGFP) and mCherry probes at position 1591 in the WT and Chim subunit, respectively. The presence of fluorescent proteins at this position does not interfere with the mechanical sensitivity of the channels measured electrophysiologically (Fig. 4a)[38]. Confocal images obtained from cells transfected with various ratiometric plasmid mixtures show that the ratio of fluorescence emission of mCherry over eGFP is directly proportional to the transfected plasmid mixture (Fig. 4b–d). In addition, the green and red emissions are strongly co-localized (Fig. 4e), indicating that the WT and Chim subunits interact. Another needed control is to verify that, in a mixed population of channels, the $\Delta F/F_0$ induced by Yoda1 is proportional to the relative fraction of Yoda1-sensitive channels. To this aim, we transfected different ratios of plasmids encoding mPZ1 and mPZ2 and measured the corresponding Yoda1 dose–response. Our data show that the $\Delta F/F_0$ values relative to the $\Delta F/F_0$ obtained from a transfected mixture containing 100% mPZ1 (relative $\Delta F/F_0$) is proportional to the relative fraction of mPZ1 contained in the transfected plasmid mixture (Fig. 4f, g). Therefore, the measured Yoda1 dose–response is proportional to the relative fraction of Yoda1-sensitive channels present in the cells.

Assuming channels are formed by the random assembly of Chim and WT subunits (random mixing), the fraction (Fr) of trimeric channels with $i$ WT subunits is given by the equation:

$$\text{Fr}_{(i)} = \binom{3}{i} f_{Ch}^{(3-i)} f_{WT}^{i} \qquad (5)$$

with $f_{Ch}$ and $f_{WT}$ the relative fractions of transfected plasmids encoding the chimeric and WT subunits, respectively. If WT and Chim subunits were unable to co-assemble (no mixing), the $\text{Fr}_{(i)}$ values would be either zero (for $i = 1$ and $i = 2$) or equal to $f_{WT}$ for $i = 3$. Based on Equations 2 and 5, the fluorescence signal from a heterogeneous population of hybrid channels containing $i$ WT subunit(s) will be a function of the saturation fraction by the agonist multiplied by the sum of the relative fractions of channel species contributing to the signal:

$$\frac{\Delta F}{F_0} = B_{max}\left(\frac{[L]}{[L] + K_d}\right)\left(\alpha \text{Fr}_{(i=3)} + \beta \text{Fr}_{(i=2)} + \gamma \text{Fr}_{(i=1)}\right) \qquad (6)$$

with $\alpha$, $\beta$, and $\gamma$ coefficients with a value of 1 or 0. For instance, if one assumes channels with 1 or more WT subunits contribute to the Ca$^{2+}$-induced fluorescence signal, all three coefficients are equal to one. If one assumes only channels with 2 WT subunits contribute to the Ca$^{2+}$-induced fluorescence signal, then $\alpha = \gamma = 0$ and $\beta = 1$. For simplicity, the numeral term corresponding to the number of subunit in the numerator in Eq. 2 has been eliminated and included in $B_{max}$.

We characterized Yoda1-induced calcium responses of cells expressing four different Chim:WT ratios, ranging from 100% WT (Chim:WT = 0:1) to 10% WT (Chim:WT = 9:1). The relative distribution of channel species assuming random mixing (i.e., calculated using Eq. 5) is shown in Fig. 5a. A global fit of the data assuming no mixing of the subunits yielded a poor determination coefficient ($R^2 = 0.808$, $n = 4$) (Fig. 5b and Table 1). Poor global fits were also obtained assuming random mixing and assuming that the contributing channel species contain either three WT subunits (3 ATM) (Fig. 5c and Table 1) or more than 2 WT subunits (≥2 ATM) (Fig. 5d and Table 1). In contrast, good global fits were obtained assuming random mixing and a contributing fraction made of channels containing at least one WT subunit (≥1 ATM) (Fig. 5e and Table 1).

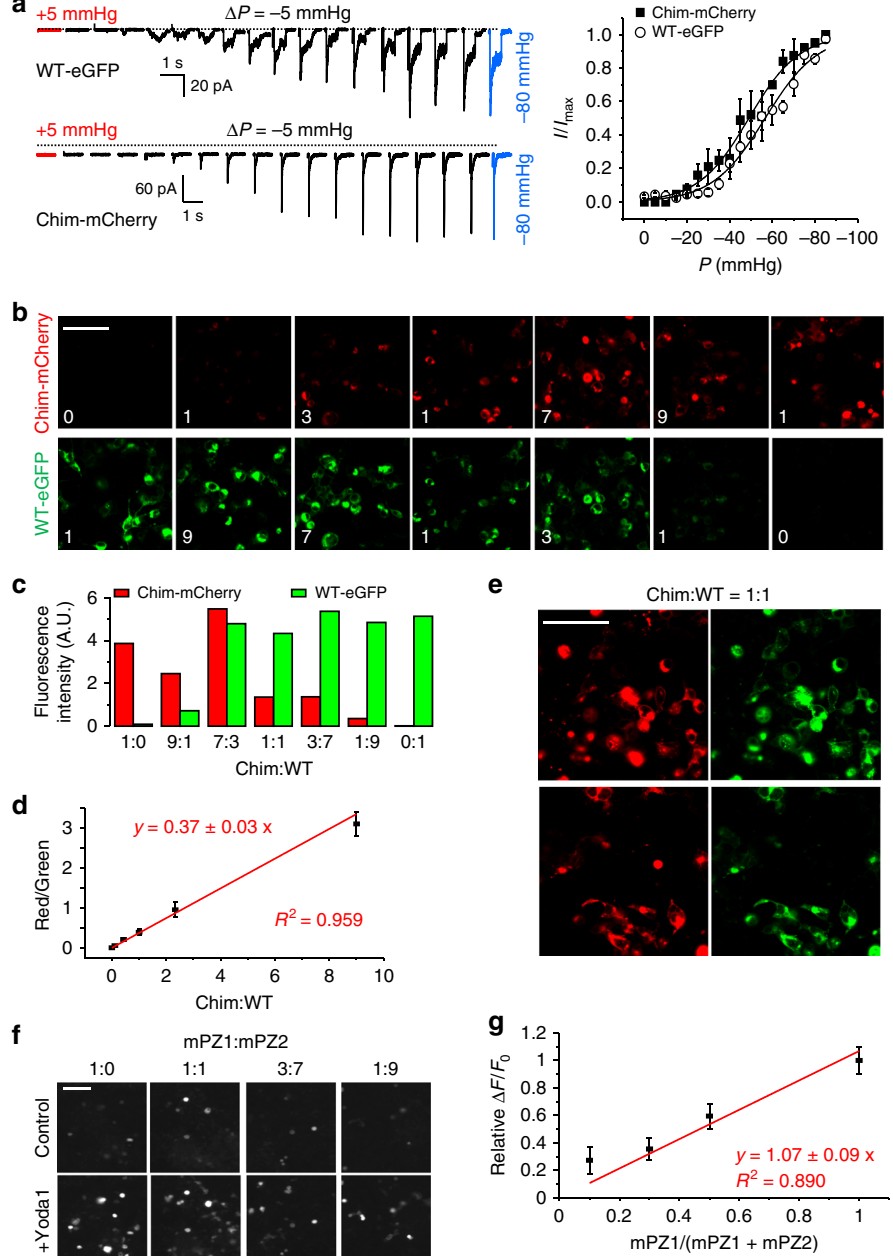

**Fig. 4** Ratiometric expression of hybrid channels. **a** Left, pressure-induced ionic current traces from WT-eGFP and Chim-mCherry. Right, normalized peak current plotted against pressure for WT-eGFP (open circles) and Chim-mCherry (full squares). The plots are the average of at least five independent recordings. Dotted lines indicate zero current level. **b** Example of confocal fluorescence images showing separate eGFP emission (bottom row) and mCherry emission (top row) from HEK 293T cells co-transfected with WT-eGFP and Chim-mCherry. The number in each image indicates the relative proportion of each transfected plasmid. **c** Histograms showing the fluorescence intensity from images displayed in **b** for different ratiometric transfections. **d** Linear fitting of the relative mCherry/eGFP emission (red/green) as a function of the relative proportion of Chim/WT transfected plasmids (Chim:WT). The experiment was repeated three times. **e** Examples of confocal images from cells transfected with equimolar amounts of plasmids encoding WT-eGFP and Chim-mCherry. The dynamic range of the images has been altered to highlight the co-localization of both fluorescence channels. **f** Examples of calcium-sensitive fluorescence images obtained using GC6 before (control) or after incubation with 30 µM Yoda1 (+Yoda1) in cells transfected with the indicated ratios of plasmids encoding mPZ1 and mPZ2. **g** The $\Delta F/F_0$ obtained for each ratio tested in **f** was normalized to the $\Delta F/F_0$ obtained in cells transfected only with mPZ1 (relative $\Delta F/F_0$) and plotted as a function of the proportion of mPZ1 plasmid in the plasmid mixture (mPZ1/ (mPZ1 + mPZ2)). A linear fit to the data is shown in red. The experiment was done in duplicate and repeated four times. In **a**, **d**, **g**, error bars = s.e.m. Scale bar in **b**, **e**, **f** = 50 µm

Another way to determine the minimal Yoda1-sensitive subunit stoichiometry required for chemical gating is to compare observed vs. predicted fraction of channel species contributing to the fluorescence signal. Since the contributing fraction of WT channels is equal to 1, we can determine the contributing fraction of various hybrid channel mixtures by taking the ratios of the $\Delta F$/

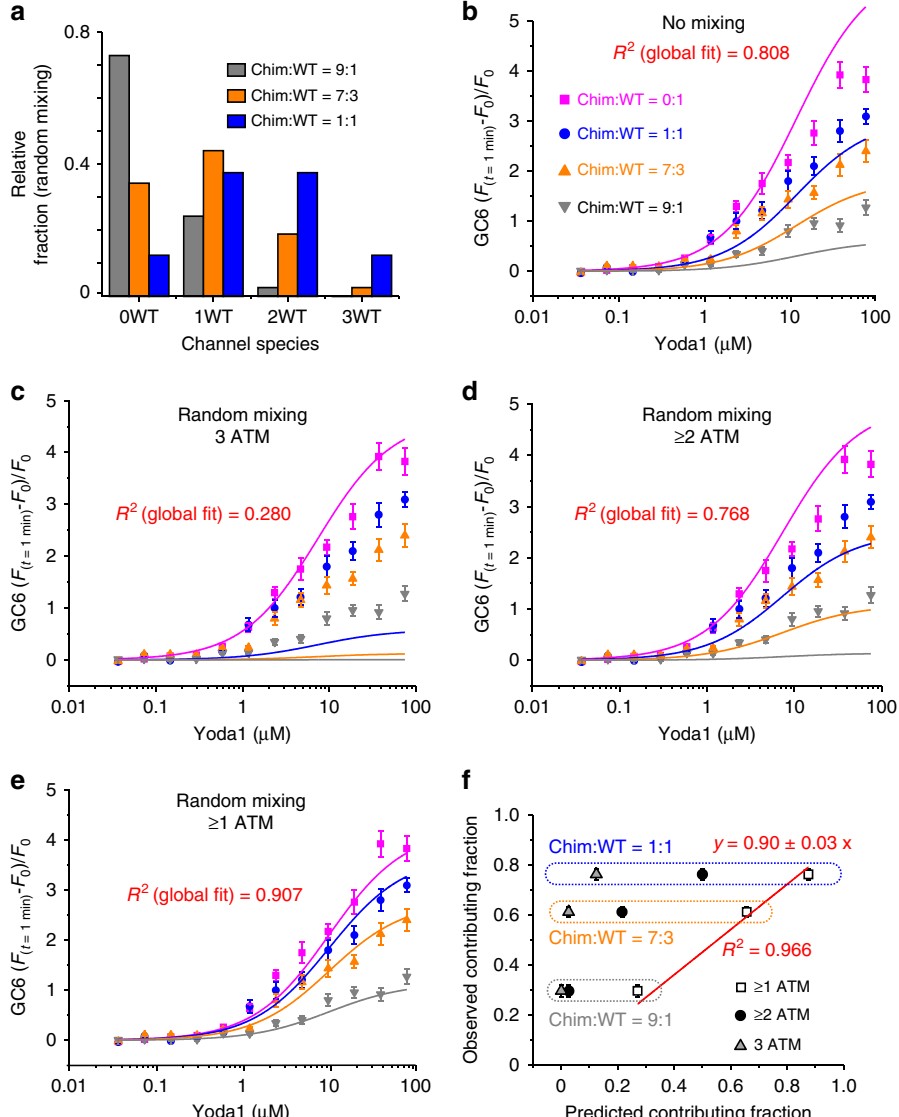

**Fig. 5** Global fitting of Yoda1 dose–responses from hybrid channels. **a** Predicted distribution of channels species for the indicated Chim:WT ratios calculated using Eq. 5. **b**–**e** Yoda1 dose–responses from cells transfected with the indicated Chim:WT ratios (0:1, magenta; 1:1, blue; 7:3, orange and 9:1, dark gray). The experiment was done in duplicate and repeated five times. The data were fitted with four different gating models using Equation 6. In **b**, the model assumes no subunit mixing, thus the fluorescence signal is predicted to be contributed by WT channels whose relative proportion is identical to the proportion of transfected WT plasmid. In **c**–**e**, the models assume random subunit mixing with relative distributions as depicted in **a**. These models predict that the fluorescence signal is contributed by channels harboring 3 WT subunits (3 ATM) (**c**), or at least two WT subunits (≥2 ATM) (**d**) or at least one WT subunit (≥1 ATM) (**e**) (see Table 1 for more details). **f** The observed contributing fraction of Yoda1-activated channels is plotted against the predicted contributing fraction of Yoda1-activated channels using Equation 7 and for three different Chim:WT ratios (1:1, blue; 7:3, orange and 9:1, gray). The predicted fractions were calculated assuming random subunit mixing and assuming signal contribution by channels having 3 ATM (gray triangles), at least 2 ATM (black circles) or at least 1 ATM (open squares). In **b**–**f**, error bars = s.e.m

$F_0$ between hybrid (mix) and WT channels:

$$\frac{\frac{\Delta F}{F_0}(\text{mix})}{\frac{\Delta F}{F_0}(\text{WT})} = \text{observed contributing fraction} \quad (7)$$

With $\Delta F/F_{0(\text{mix})}$ the calcium fluorescence signal observed when mixing WT and chimeric subunits and $\Delta F/F_{0(\text{WT})}$ the calcium fluorescence signal observed for WT channels. We determined the observed contributing fraction by calculating this ratio for each Ch:WT ratio. This number was averaged from the observed $\Delta F/F_0$ obtained for six tested Yoda1 concentrations ranging between 2 and 75 μM. We next calculated the predicted

contributing fraction for three Chim:WT ratios (i.e., 1:1, 7:3 and 1:9) by calculated the sum on the right of Equation 6 with the assumptions that the observed signal is contributed by channels containing more than one ($\alpha = \beta = \gamma = 1$), more than 2 ($\alpha = 0$; $\beta = \gamma = 1$) or 3 WT subunits ($\alpha = \beta = 0$; $\gamma = 1$) (Fig. 5f). The observed contributing fractions were then plotted against their corresponding predicted contributing fractions. A linear fit of the data shows that the observed fractions correlate very well with the predicted fractions assuming channels with at least one WT subunit are contributor ($R^2 = 0.966$, Fig. 5f). Thus, our data strongly indicate that heterotrimeric channels possessing only a single WT subunit exhibit Yoda1 sensitivity similar to WT channels.

**Table 1 Global fit results for different gating models**

| Number of contributing subunit(s) | ≥1 ($\alpha = \beta = \gamma = 1$) (random mixing) | ≥2 ($\alpha = 0; \beta = \gamma = 1$) (random mixing) | 3 ($\alpha = \beta = 0; \gamma = 1$) (random mixing) | 3 ($\alpha = \beta = 0; \gamma = 1$) (no mixing) |
|---|---|---|---|---|
| $B_{max}$ | 4.21 ± 0.32 | 5.00 ± 0.48 | 4.68 ± 0.96 | 6.11 ± 0.75 |
| $K_d$ | 9.62 ± 1.41 | 7.33 ± 2.35 | 7.40 ± 5.05 | 11.74 ± 2.67 |
| $R^2$ | 0.907 | 0.768 | 0.280 | 0.808 |

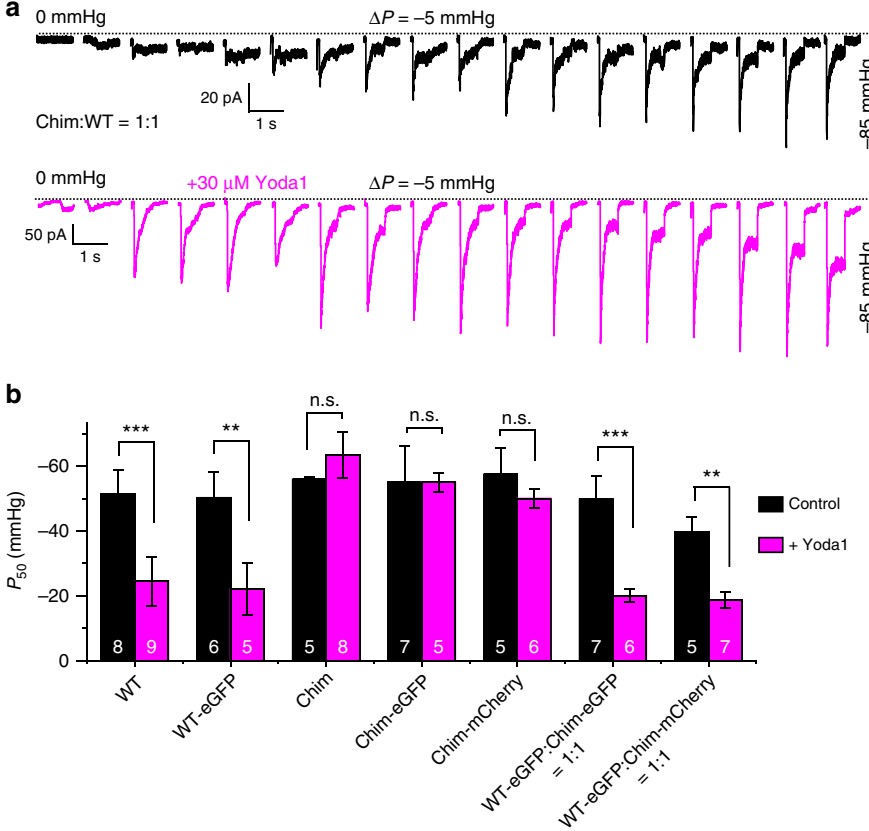

**Fig. 6** Yoda1 activates Chim:WT hybrid channels similarly to WT Piezo1. **a** Example of pressure-induced ionic current recordings in cell-attached patches from cells transfected with an equimolar mixture of WT and Chim plasmids. Dotted lines indicate zero current level. **b** Histograms showing $P_{50}$ values for the indicated Piezo1 constructs in the presence (magenta bars) or absence (black bars) of 30 μM Yoda1. The effect of Yoda1 on $P_{50}$ was tested for each construct using a student $t$-test. Asterisks indicate standard $p$-value range: **: $0.001 < p < 0.01$; ***: $p < 0.001$; n.s.: non-significant. In **a** and **b**, $V = -80$ mV. In **b**, error bars = s.e.m. and the numbers in each bar indicate the number of recordings analyzed

If hybrid channels exhibit Yoda1-induced calcium response like WT mPZ1, they should also exhibit similar Yoda1-induced changes in mechanosensitivity. To test this hypothesis, we performed on-cell pressure-clamp electrophysiology recordings with cells transfected with equimolar ratios of WT and Chim plasmids. In these conditions, Equation 5 predicts a distribution of 75% hybrid channels, 12.5% of WT channels and 12.5% of Yoda1-insensitive Chim channels. Hence, a large majority of stretch-induced currents from the patches are expected to be from hybrid channels. Figure 6a shows that the presence of Yoda1 reduces the mechanical threshold for channel activation, similar to WT mPZ1. We also determined the $P_{50}$ values of WT, Chim and hybrid channels harboring or not a fluorescent protein eGFP or mCherry (Fig. 6b). We found that, regardless of the presence of a fluorescent protein, application of 30 μM Yoda1 shifts $P_{50}$

values by +20 to +30 mmHg in hybrid channels ($t$-test $p$-values ranging from 0.002 to 0.007) and WT channels ($t$-test $p$-values ranging from 0.0009 to 0.005) but not in Chim channels ($t$-test $p$-values ranging from 0.3 to 0.9).

To compare more quantitatively the effect of Yoda1 between WT and hybrid channels, we measured the $I/I_{max}$ vs. pressure plots in the presence of varying Yoda1 concentrations in cell-attached patches from cells transfected with WT or with an equimolar mixture of WT and Chim plasmid (Fig. 7). Since Yoda1 was applied through the patch pipette, each Yoda1 concentration was tested on separate patches. Interestingly, in presence of 100 μM Yoda1 concentration, the activation slope $k$ is higher (6.9 ± 0.6 mmHg, $n = 4$) than at lower concentration (10.2 ± 0.7 mmHg, $n = 4$ to 10.9 ± 0.7 mmHg, $n = 4$) (Supplementary Table 3), consistent with an increased homogeneity of the

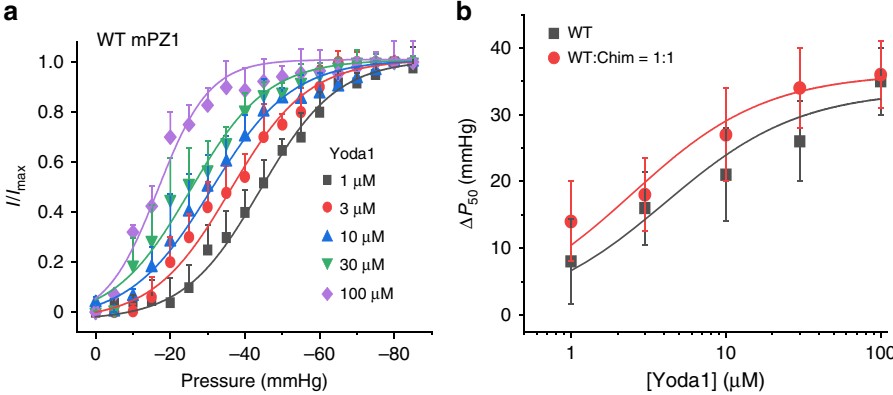

**Fig. 7** Modulation of $P_{50}$ by varying Yoda1 concentrations. **a** Normalized peak current obtained using cell-attached pressure-clamp measurements in HEK 293T cells transfected with WT mPZ1 channels and plotted as a function of the pressure pulse for the indicated Yoda1 concentrations in the patch pipette ($V = -80$ mV). **b** Plots showing the shift of $P_{50}$ values ($\Delta P_{50}$) as a function of the Yoda1 concentration measured in cells transfected with WT mPZ1 (dark gray) or with the WT:Chim = 1:1 ratio (red). The dark gray and red lines are a fit to the data using a standard binding equation similar to Eq. 2. In **a** and **b**, the plots correspond to an average of four independent experiments and error bars = s.e.m.

channel population[39]. Our data further show that increasing Yoda1 concentration reduces $P_{50}$ toward lower pressure values in both WT and hybrid channels. The plots of $\Delta P_{50}$ vs. Yoda1 concentration could be fitted using a standard binding model depicted in Eq. 2 and yielded apparent affinities that are similar between WT mPZ1 and hybrid channels ($K_d^{WT} = 4.1 \pm 2.5$ μM, $n = 4$ and $K_d^{WT:Chim=1:1} = 2.5 \pm 1.7$ μM, $n = 4$). Together with calcium imaging experiments, these results show that the shift of mechanosensitivity produced by Yoda1 is similar in mPZ1 channels containing one, two or three WT subunits.

**Hybrid channels do not produce intermediate open states**. The fact that the presence of only one Yoda1-sensitive subunit is sufficient for Yoda1-mediated activation suggests that the Yoda1-sensitive subunits may be able to partially open the pore. If true, hybrid channels would most likely possess multiple intermediate open states. We thus sought to determine the presence of such partial open states in hybrid channels using single-channel recordings in cell-attached patches. Because a small fraction of WT channel is predicted to exist when mixing WT and Chim subunits, we recorded from patches containing many channels to increase the likelihood that currents from hybrid channels are being recorded. Since strong mechanical stimuli will activate both WT and Chim subunits, we recorded single-channel's activities at low patch-pressures, i.e., at which the probability of WT subunits to be activated in the presence of Yoda1 is high but the probability of Yoda1-insensitive Chim subunits to be activated is low. Figure 5a shows examples of single channel recordings for WT and for Hybrid channels in presence of Yoda1 at the indicated pressures. The currents display characteristic steps that were further analyzed by fitting the count vs. amplitude histograms by a Gaussian function. Our data indicate that the peak-to-peak distance remains constant in presence or absence of Yoda1 for both WT and hybrid channels, indicating no change in single channel conductance (Fig. 8c, d). We did observe intermediate current levels in some recordings, but those events have extremely short dwell times compared to more stable events. Thus, the presence of the agonist Yoda1 does not create long-lived intermediate open states in hybrid or WT channels.

## Discussion

In this paper, we ask how many Piezo1 subunits need to be activated to open the channel pore. We used the small molecule agonist Yoda1 to address this question. We first identified a

minimal agonist transduction motif encompassing 103 amino acids (mPZ1 residues 1961–2063) and created an agonist-insensitive Piezo1 chimera. When aligning the amino acid sequences of human and mouse Piezo homologs, 21 residues within this identified motif are not conserved between PZ1 and PZ2 (Supplementary Figure 1), hence some of these residues may form a PZ1-specific Yoda1 binding site or they may be important to allosterically couple the binding of Yoda1 to the pore.

Yoda1 could also act by changing the lipid environment surrounding the channel protein. The toxin peptide GsMTx4, for instance, inhibits many mechanosensitive channels by affecting the lipid bilayer. However, the homolog mPiezo2, which shares ~50% of its amino acid sequence with mPiezo1, is insensitive to Yoda1. In addition, Yoda1 analogs with small chemical changes such as dichloro substitutions or oxidation of the thioether group are ineffective PZ1 agonists[35]. Recently, the Yoda1 analog Dooku1 was shown to inhibit the agonist effects of Yoda1[40], suggesting Dooku1 acts as a competitive inhibitor of Yoda1. Finally, a physical interaction between Yoda1 and mPiezo1 has been reported using surface plasmon resonance[41]. Together, these observations strongly support the existence of a proteinogenic Yoda1 binding on the Piezo1 channel. Future studies will be required to characterize the interaction between the channel and its agonist with greater details. Understanding this interaction will represent a unique opportunity to design selective modulators against Piezo channels.

Next, we sought to create heteromeric channels by transfecting cells with various mixtures of plasmids encoding WT and chimeric subunits. This approach has been extensively applied to probe the gating mechanism of multimeric ion channels by endogenous stimuli and toxins[42–44]. However, this approach does not allow us to control the subunit composition of the expressed channels. Ideally, this could be done by covalently linking the channel subunits. Although this method has been successfully applied to smaller multimeric proteins such as potassium channels, it is presently unclear whether the genetic linkage of the much larger Piezo subunits would be technically achievable and that it would yield functional channels. Instead, our study employs a careful fitting analysis to determine the minimal stoichiometry of Yoda1-sensitive subunits needed to confer Yoda1 sensitivity to the Piezo1 channel. Our data demonstrate that Yoda1 activate hybrid channels containing as few as one Yoda1-sensitive subunits. The effects of the agonist on these hybrid channels were moreover very similar to homotrimeric WT channels. Since the pore of Piezo channels is constituted by

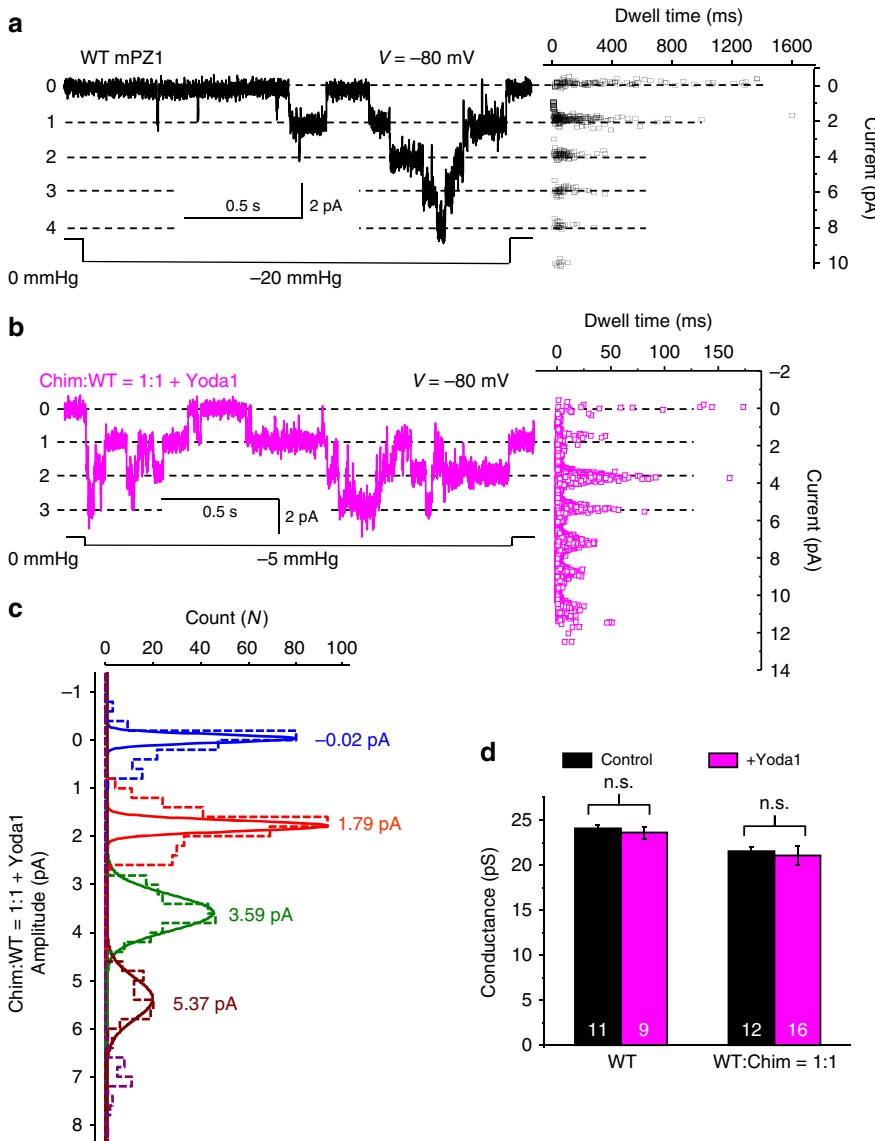

**Fig. 8** Yoda1 does not produce long-lived intermediate open states. **a** Left: example of pressure-evoked single channel currents recorded in cell-attached patches from WT mPZ1 channels at −80 mV. Right: Plot showing the dwell time vs. current amplitude of detected events. **b** Same as **a** for hybrid channels assembled from an equimolar amount of Chim and WT subunits (V = −80 mV). **c** Mean amplitude of single channel currents determined by a Gaussian fit of the amplitude distribution of single channel currents from cell-attached patches expressing hybrid channels (blue: no open channel, red: one open channel, green: 2 open channels and brown: 3 open channels). **d** Histograms showing single channel conductance measured at −80 mV for WT and hybrid channels (Chim:WT = 1:1) in the presence (magenta) or absence (black) of 30 μM Yoda1. The effect of Yoda1 on single channel conductance was tested using a student *t*-test. n.s.: non-significant. In **a–c**, dwell time and mean amplitude were obtained by averaging at least 20 current traces obtained from at least six independent cell-attached patches. In **d**, error bars = s.e.m. and the numbers in each bar indicate the number of independent traces analyzed

amino acids from all three subunits, our observations are theoretically consistent with two gating mechanisms. First, the pore could be gated by a concerted transition in all three subunits. In this case, locking one subunit in its activated state with the agonist would automatically lock the other coupled subunits in their activated conformation, thus maintaining a fully conducting pore (Fig. 9a). On the other hand, the pore could be gated by independent transitions in each subunit wherein activation of a single subunit is sufficient to produce a fully open pore (Fig. 9b).

Most homomultimeric ion channels harboring a single pore like Piezo1 seem to open in a switch-like fashion by a concerted mechanism[45–49] while only few examples of sequential gating have been reported[50]. In addition, sequential gating is usually associated with permeation properties that increase incrementally as more subunits are being activated. However, this does not seem to be the case for Piezo1, since we did not observe long-lived intermediate open states in hybrid channels in the presence of Yoda1. These observations tend to favor a concerted mechanism for Piezo1.

In summary, we have identified essential molecular determinants responsible for the isoform selectivity of Yoda1 between Piezo1 and Piezo2 and identified mechanistic principles by which the subunits control the channel pore. This work sheds light on the inner workings of Piezo channels and will also help the rationale design of selective pharmacological agents targeting mammalian Piezo homologs.

## Methods
**Molecular cloning**. A pCDNA3 plasmid containing mPiezo1 was obtained from Dr. Mikhail Shapiro (Caltech) and was originally a gift from Dr. Patapoutian (The

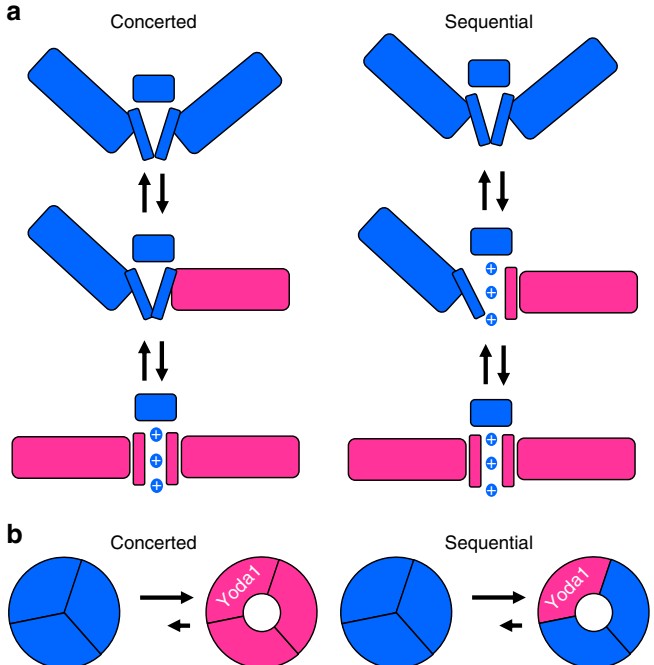

**Fig. 9** Two possible gating mechanisms of Piezo1. Our study reveals Yoda1 sensitivity of Piezo1 requires the presence of only one Yoda1-sensitive subunit per trimer. This result is consistent with a concerted gating mechanism (**a**) where all subunits cooperatively gate the pore (blue: resting subunits; magenta: activated subunits). This result is also consistent with a sequential mechanism (**b**) where the pore fully opens if only one subunit is activated

Scripps Research Institute). A Sport6 plasmid encoding mPiezo2 was directly obtained from Dr. Patapoutian. To create the polycistronic vector pCDNA3-mPZ1-IRES-GC6, we PCR-amplified the pCDNA-mPZ1 plasmid, the internal ribosome entry site (IRES) cassette from a pIRES-eGFP plasmid and the GC6 cDNA from a pGP-CMV-GC6 plasmid (Addgene plasmid #40754) and assembled them using the NEBuilder HiFi DNA Assembly kit (New England Biolabs). A similar procedure was used to create pCDNA3-mPZ2-IRES-GC6, pCDNA3-GC6, and the empty control vector pCDNA3. All mPZ1-mPZ2 chimeras with or without the IRES-GC6 cassette and with or without the fluorescent proteins eGFP and mCherry inserted at mPZ1 position 1591 were created using the same approach. All constructs were verified by automated Sanger sequencing (Genewiz).

**Cell culture and transfection.** HEK 293T cells (ATCC) were cultured in standard conditions (37 °C, 5% $CO_2$) in a DMEM medium supplemented with Penicillin (100 U mL$^{-1}$), streptomycin (0.1 mg mL$^{-1}$), 10% sterile Fetal Bovine Serum and 1×MEM non-essential amino acid without L-glutamine. All cell culture products were purchased from Sigma-Aldrich. Transfection was done on cells with a passage number lower than 25 using polyethylenimine (PEI, Polysciences #23966). Briefly, a sterile mixture of DNA:PEI (1:4 weight ratio) was prepared using sterile 100 mM NaCl solution and added directly to the cell's culture medium (220 ng of total DNA was added per cm$^2$ of cultured surface). Culture medium was changed 16–20 h after transfection. For fluorescence experiments, cells seeded on clear-bottom black 96-well plates (Nunc) were transfected at ~50% confluence 2–4 days before the experiment. For electrophysiology experiments, cells seeded on uncoated coverslips were transfected at ~10% confluence 12–36 hours prior recordings.

**Calcium imaging, osmotic shocks, and chemical treatment.** Fluo8 AM was purchased from abcam (ab142773), dissolved in dimethyl sulfoxide (DMSO) and stored in 5 mM aliquots at -20 °C. Yoda1 was purchased from Sigma-Aldrich (#SML1558), dissolved in DMSO and stored in 10 mM aliquots at −20 °C. For experiments with Fluo8-AM, cells were washed with a isotonic (300–310 mOsm L$^{-1}$) normal physiological saline solution (NPSS) containing 140 mM NaCl, 5 mM KCl, 2 mM MgCl$_2$, 1 mM CaCl$_2$, 10 mM HEPES (pH 7.4 with HCl or NaOH) and 10 mM Glucose and incubated for 1 h in a 37 °C/5% $CO_2$ incubator with a NPSS solution containing 3–5 μM Fluo8 AM. After incubation, cells were washed again with 100 μL NPSS and placed on a Nikon inverted fluorescence microscope. Epi-fluorescence excitation was provided by a 100 W mercury lamp through a ×20 objective and fluorescence images were obtained using a standard GFP filter set and acquired at 1 frame/s using a Nikon Digital Sight camera and the Nikon Digital

Element D software. During recordings, 100 μL of a 2×Yoda1 NPSS solution was added to the cells at $t = 30$ s or sometimes at $t = 10$ s. Total DMSO concentration was kept below 1% for all tested Yoda concentrations. For osmotic shocks, 200 μL of a hypotonic solution (40–50 mOsm L$^{-1}$) was added to the 100 μL isotonic NPSS during the recording. Hypotonic solution was identical to NPSS except the NaCl concentration was reduced from 140 to 5 mM. For experiments with GC6, fluorescence imaging was done similarly except the 1 h Fluo8 AM treatment was replaced by a 1 h NPSS incubation. Fluorescence images were analyzed with ImageJ. Each experiment was done in duplicate and repeated n times (n values are indicated in each figure). For each experiment, the fluorescence intensity taken from 30 cells (15 cells per image stack in duplicate) was averaged. Cells were selected randomly for analysis at the beginning of each image acquisition before Yoda1 application. Dead cells with abnormal shape or unusual high fluorescence were excluded from analyses. The final $F_{(t=1min)} − F_0/F_0$ ($\Delta F/F_0$) values and standard errors were calculated by averaging the mean values from n experiment. Fitting Yoda1 dose–responses were done using OriginPro 2017.

**Confocal imaging.** Cells were imaged 2–3 days after transfection on a Zeiss LSM 880 fluorescence confocal microscope using the Zen acquisition software. Fluorescence emission from eGFP and mCherry were collected using dedicated green and red channels and with a ×63 lens. All image parameters were kept constant when comparing different transfection ratios. Unless otherwise stated, the images were unmodified.

**Cell-attached patch-clamp recordings.** During recordings, the membrane potential was zeroed using a depolarizing bath solution containing 140 mM KCl, 1 mM MgCl$_2$, 10 mM glucose and 10 mM HEPES pH 7.3 (with KOH). Fire-polished patch pipettes with a diameter of 2–3 μm and resistance of 1–5 MΩ were filled with a recording solution containing 130 mM NaCl, 5 mM KCl, 1 mM CaCl$_2$, 1 mM MgCl$_2$, 10 mM TEA-Cl and 10 mM HEPES pH 7.3 (with NaOH). Stretch-activated currents were recorded in the cell-attached configuration after seal formation using a Multiclamp 700B or Axopatch 200B amplifiers (Axon) and a high-speed pressure clamp (HSPC-1, ALA Scientific Instruments). The membrane potential inside the patch was held at −80 mV. Data were recorded at a sampling frequency of 10 kHz and filtered offline using a 1–2 kHz lowpass digital filter and a digital filter to remove electrical interferences (pClamp, Axon). Peak current amplitudes were determined by adjusting the baseline prior the pressure pulse. The relative peak current values and standard error are from averaging recordings from n transfected cells. No mechanosensitive ionic currents above baseline could be detected in non-transfected HEK 293T cells. Fitting the plots of $I/I_{max}$ vs. pressure was done using Eq. 1 in OriginPro 2017.

**Single-channel analysis.** Traces showing single channel events were filtered at 1 kHz and analyzed using Clampfit (Axon). Events shorter than 0.5 ms were excluded during automatic event detection. We plotted both the events dwell time and events count number against current amplitudes. The current amplitude corresponding to the main open states were determined by fitting each peak of the count vs. amplitude plots by a Gaussian function (Clampfit). The conductance was calculated by averaging the peak-to-peak amplitude.

**Statistics.** To test statistical difference between two mean values, we performed two-tailed unpaired Student's t-tests. The standard p-value range is indicated in each figure.

**Data availability.** All data are available from the corresponding author upon reasonable request.

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

## Acknowledgements

We thank Dr. Xiaoning Bi for lending us her inverted fluorescence microscope, Dr. Mikhail Shapiro from lending us his Multiclamp 700B amplifier, and Dr. Ardem Pata-poutian for his gift of the mPiezo1 & 2 cDNAs. We also thank Andrew Tran, Teagen Vence, Jerry Harb, Rachel Chang, and Rachel Abraham for technical help with molecular biology. This work was funded by start-up funds from the Western University of Health Sciences (to J.J.L. and Y.L.).

## Author contributions

J.J.L. conceived the project, performed experiments, analyzed data and wrote the paper. J.J.L., W.M.B.-S. and Y.L. derived equations.

## Additional information

**Competing interests:** The authors declare no competing interests.

