## [Peer Review File · Nature Communications]

Reviewers' comments:

Reviewer #1 (Remarks to the Author):

This paper studies the mechanism of action of the small molecule agonist Yoda1 on Piezo1. By engineering chimeras between mPiezo1 and Yoda1-insensitive mPiezo2 the authors identify a minimal agonist binding/transduction motif (aa 1961-2063) on Piezo1 required for Yoda1-sensitivity. The Piezo1 region is located at the interface between the blade and pore domains in each subunit. They further investigated the effect of Yoda1 on heterotrimeric Piezo1 channels harboring WT subunits (which bind Yoda1) and Yoda1-insensitive mutant subunits. They show that hybrid channels harboring as few as one Yoda1-sensitive subunit exhibit Yoda1-sensitivity similar to homotrimeric wild type channels, suggesting that binding of Yoda1 to a single Piezo1 subunit is needed to open the channel pore. The structural mechanism by which Yoda1 interacts with Piezo1 was not addressed.

Overall it is a good, well executed work, with well designed experiments, but that does not fully address the question. More precise piezo 1 mutations would have been required to better delineate key residues and provide information about how activation of a single subunit translates to the multimeric pore.

Major concerns:

1) To investigate the number of Yoda1 molecules that binds/activates Piezo 1, the authors used calcium imaging, which is not a decisive technique. Since intracellular calcium concentration increases when mpiezo1-expressing cells were exposed to Yoda1 (i.e. at basal membrane tension), why not recording inward currents generated with increasing concentrations of Yoda1 (e.g. without mechanical stimulation). And then perform curve fitting of Yoda1 dose-responses using equations that describe the interaction of a ligand with multiple protein binding sites.

2) The chimeras were functionally tested based on their ability to evoke calcium signals in response to membrane stretch induced by an acute hypotonic shock. It's ok, but it would be better to test mechanosensitivity by using pressure clamp rather than hypotonic shock. In Fig 2, it seems that the chimera shows a rightward shift in mechanosensitivity. Any comments?

3) Channel opening is evoked using brief negative pressure pulses applied to the backside of the patch pipette while maintaining a patch potential of -80 mV relative the inside of the cell. OK. But it is important to indicate in the figures the on and off of the pressure stimulus (Fig 7a,b). The same hold with the application of Yoda1 (Fig1c).

4) What is the unitary conductance of the channel activated by Yoda1 at resting membrane tension (no pressure applied in the patch pipette)?

Minor points:

Page 7: able to sojourn into its open state

Page 8. Ch1961-2063 (from now on named "Chim") which exhibit an apparent loss of Yoda1-sensitivity

Page 8. a patch potential of -80 mV relative the inside of the cell
Chim+Yoda1 Figure 2c, wrong symbols

Methods: Total DMSO concentration was kept below 1% for all tested Yoda concentrations. Sounds huge! No effects of such a concentration?

Reviewer #2 (Remarks to the Author):

L28. "Yoda1, a Piezo1-selective small molecule agonist, is 28 the only known selective Piezo modulator." Not true. Gsmtx4 is an excellent modulator. Which channels have been checked for a kinetic response to Yoda1 to say it is specific for Piezo?

L49 "An avenue to treat these diseases would be to correct Piezo channel activity with small 50 molecule agonists and/or antagonists." But you just said the channels are everywhere so what could be the consequence of giving something so general?

L136+ put derivation in supplement

How do you know that yoda does not bind to the bilayer and thereby alters stress on the channel?

How are you accounting for the internal kinetics of Ca²⁺ handling? That affects kinetic analysis.

The title is misleading. Something like "YODA1 interaction with regions of Piezo1" is better

My pdf of the paper does not display videos

Reviewer #3 (Remarks to the Author):

I find the paper "Probing the Gating Mechanism of the Mechanosensitive Channel Piezo1 with a Small Molecule Agonist" very informative. The experiments have been carefully done and presented. The paper is written thoughtfully.

By bringing in Yoda1, a small activator molecule, the authors go beyond purely tension-dependent aspects of Piezo1 gating and probe the possible cooperativity and allostery in its mechanism.

General:

The allosteric effect of Yoda1 binding on the open-state energy should include contributions from all subunits, whether they gate cooperatively or not. The authors may consider three additional experiments.

(1) Because the energy difference between the open and closed states is directly reflected in P50, it would be good to see the dependence of P50 on Yoda1 concentration, covering the entire range below and above the k_d (~10 μ m).

(2) If the authors took careful measurements of P_o versus pressure (tension) near k_d (10 μm), they expect to see the response of a mixed population with different occupancy by the drug. Not only P_{50} should be different from the saturating Yoda concentration, but the slope factor k should be lower (see the reasoning in BJ 86 (2004) 2846).

(3) When the measurement is taken on WT:Chim 1:1 population, the slope factor k even at saturating Yoda concentrations should be measurably lower compared to either WT or Chim uniform populations, for the same reason of non-uniformity. It would be surprising to see if this is not the case.

I have several more particular suggestions that may improve readability of this paper.

Fig. 1. The vertical scale in panels d and e is the same, but the applied stimuli are different. In the first case ch1961 is totally inactive, and fully active in the latter. For the ease of figure readability it would be desirable to put the words 'osmotic activation' on panel d.

Please correct symbols on Fig. 2c.

The results presented on panels f and g are not discussed in detail further in the paper, yet they are interesting. They suggest that the binding site immediately coordinating Yoda1 I is likely formed by the C-terminal domain apparently involving Ch1 segment, which binds the drug initially with a higher affinity, but fails to produce full activation. The 'normal' activating effect requires a larger chunk of protein, imposing lower affinity at low saturations, highly consistent with the allosteric mechanism.

It would be desirable to mention the magnitude of osmotic drop in the shock experiments right in the text.

Line 123: I would suggest putting: '...in the absence of external mechanical stimuli...' because we don't know the distribution of membrane tension at rest.

Lines 126-132: please mention whether these were on-cell or excised patches and the pipette size.

Line 136, eqn 1: is the midpoint designated as $P_{1/2}$ or P_{50} ? Choose one.

Line 136: ' $I = I_{\text{max}}/2$ and the slope factor k , a constant expressed in mmHg, that'

Line 178: '... making it very difficult to link...' – the authors may also mention the problem that the C- and N- termini may not be close by thus precluding concatenation without a long unnatural linker.

Lines 183-185 and all electrophysiology figures: The authors may comment on the difference in channel adaptation times between WT and Chim.

Line 209: '...WT subunits contribute to the Ca^{2+} -induced fluorescence signal...'

Line 285: '... some of the residues may form a classical Yoda1 binding site...'

Line 322: the

Responses to the reviewers

For clarity, our response are indicated in blue.

Reviewer #1 (Remarks to the Author):

This paper studies the mechanism of action of the small molecule agonist Yoda1 on Piezo1. By engineering chimeras between mPiezo1 and Yoda1-insensitive mPiezo2 the authors identify a minimal agonist binding/transduction motif (aa 1961-2063) on Piezo1 required for Yoda1-sensitivity. The Piezo1 region is located at the interface between the blade and pore domains in each subunit. They further investigated the effect of Yoda1 on heterotrimeric Piezo1 channels harboring WT subunits (which bind Yoda1) and Yoda1-insensitive mutant subunits. They show that hybrid channels harboring as few as one Yoda1-sensitive subunit exhibit Yoda1-sensitivity similar to homotrimeric wild type channels, suggesting that binding of Yoda1 to a single Piezo1 subunit is needed to open the channel pore. The structural mechanism by which Yoda1 interacts with Piezo1 was not addressed.

Overall it is a good, well executed work, with well-designed experiments, but that does not fully address the question. More precise piezo 1 mutations would have been required to better delineate key residues and provide information about how activation of a single subunit translates to the multimeric pore.

We are particularly thankful to reviewer#1 for his overall positive comment. We agree that the structural mechanisms by which Yoda1 stabilizes the channel subunits in the open state have not been addressed. However, this work was not focused on addressing the structural mechanism of Yoda1-mediated channel activation, which would require a different set of experimental and computational approaches and thus would be beyond the scope of this study. Our goal here was to identify a Yoda1-insensitive mutant and to use that mutant to address one important question: how many agonist-sensitive subunits are needed to mediate agonist activation of Piezo1? We believe our work has addressed this question.

We acknowledge that the exact mechanism by which the pore opens is unclear. Our data support two hypotheses: one is the classical concerted transition wherein all subunits cooperatively open the pore. The other is a mechanism wherein independent motions of each subunit are sufficient to create an open pore but whose conductance would not further increase with incremental activation of the remaining subunits.

Many homomultimeric channels like Piezo open their pore in a concerted fashion. However, recent cryo-EM mPiezo1 structures derived from symmetry-free classification revealed that each subunit within the trimeric complex is able to move independently and such independent motion might be functionally relevant, i.e., sensing the unevenly distributed force in the membrane (Zhao et al., Nature 2017 & Saotome et al. Nature 2018). However, independent motion of sensory domains may not necessary lead to independent pore opening events by each subunit. In many homotetrameric voltage-gated potassium channels, for instance, voltage-sensing modules move largely independently until all activated modules undergo an ultimate cooperative transition that opens the channel pore. Further studies will be needed to elucidate in greater details how the three subunits operate the gate of the channel pore.

Major concerns:

1) To investigate the number of Yoda1 molecules that binds/activates Piezo 1, the authors used calcium imaging, which is not a decisive technique. Since intracellular calcium concentration

increases when mPiezo1-expressing cells were exposed to Yoda1 (i.e. at basal membrane tension), why not recording inward currents generated with increasing concentrations of Yoda1 (e.g. without mechanical stimulation). And then perform curve fitting of Yoda1 dose-responses using equations that describe the interaction of a ligand with multiple protein binding sites.

We agree that calcium imaging may not be as precise as electrophysiology recordings. The idea of recording whole-cell currents elicited upon bath application of Yoda1 in absence of mechanical stimulations has been tested by Syeda et al. Unfortunately, they were unsuccessful in detecting any measurable current in the presence of 10 micromolar Yoda1 (which approximately corresponds to the apparent EC50 of Yoda1). Hence, even if such currents could be measured above 10 μ M, curve fitting would be extremely poor because all the points below EC50 would be missing. The main reason for this has to do with the fact that Yoda1 only partially activates Piezo1, increasing open probability only slightly in the absence of external mechanical stimuli.

2) The chimeras were functionally tested based on their ability to evoke calcium signals in response to membrane stretch induced by an acute hypotonic shock. It's ok, but it would be better to test mechanosensitivity by using pressure clamp rather than hypotonic shock. In Fig 2, it seems that the chimera shows a rightward shift in mechanosensitivity. Any comments?

We indeed tested our chimeras using calcium imaging and tested our best candidate using hypotonic shocks only as initial characterization. We did perform careful pressure-clamp electrophysiology measurements in cell-attached patch recordings on this chimera (Fig 2). The main question we addressed was to test whether Yoda1 would change the pressure-sensitivity of the chimera. The right shift indicates the chimera is slightly less efficient in transmitting mechanical force into pore opening as compared to wild type. The chimera changes 23 amino acids in a region presumed to couple the movement of mechanosensory machinery to the pore. Hence, it is not too surprising to observe small variations in mechanical sensitivity. We have added a short comment L147-151.

3) Channel opening is evoked using brief negative pressure pulses applied to the backside of the patch pipette while maintaining a patch potential of -80 mV relative the inside of the cell. OK. But it is important to indicate in the figures the on and off of the pressure stimulus (Fig 7a,b). The same hold with the application of Yoda1 (Fig1c).

Thank you, we have indicated the time where pressure pulses are applied on Fig 8 a-b (former Fig 7) and the Yoda1 application in Fig 1c.

4) What is the unitary conductance of the channel activated by Yoda1 at resting membrane tension (no pressure applied in the patch pipette)?

In our experiments, the application of -5 to -20 mmHg pressure through the pipette does not significantly change mPiezo1 single channel conductance in absence or in presence of 30 μ M Yoda1. In their seminal paper, Syeda and colleagues have shown mPiezo1 produces single channel current level of approximately 2 pA (Fig 2D of their paper) in absence of mechanical stimulation and in the presence or absence of the agonist (and at -80mV). These currents correspond to a conductance of about 25pS which is very similar to the conductance we report (Fig 8) and to the conductance reported for WT mPiezo1 through the literature. Hence the amount of pressure does not appear to change single channel conductance, at least in the range of pressure at which single channel activity has been recorded.

Minor points:

Page 7: able to sojourn into its open state

This seems a valid sentence.

Page 8. Ch1961-2063 (from now on named “Chim”) which exhibit an apparent loss of Yoda1-sensitivity

Thank you, we have corrected the typo. For clarity, we have also replaced this sentence by: “Ch1961-2063 (Chim) which exhibits an apparent loss of Yoda1-sensitivity”

Page 8. a patch potential of -80 mV relative the inside of the cell

Thank you, we have replaced the sentence by: “a patch potential of -80 mV relative to the inside of the cell”

Chim+Yoda1 Figure 2c, wrong symbols

Thank you, these symbols have been corrected

Methods: Total DMSO concentration was kept below 1% for all tested Yoda concentrations. Sounds huge! No effects of such a concentration?

We used the exact same procedure described in Syeda et al. and dissolved Yoda1 at a stock concentration of 10mM in DMSO. The 1% DMSO concentration was only used for the maximal tested concentration of 100 μ M. For our experiment at 30 μ M, there was 0.3% DMSO. Our supplementary videos 1 and 2 show that acute incubation of a solution containing 0.1mM Yoda1 and 1% DMSO produces strong calcium signals in cells co-expressing GCaMP6m and mPiezo1, while no significant calcium signals was produced in cells expressing only GCaMP6m. Therefore, the presence of 1% DMSO does not seem to contribute to our calcium signals, at least during the duration of our assay (30 sec).

Reviewer #2 (Remarks to the Author):

L28. “Yoda1, a Piezo1-selective small molecule agonist, is 28 the only known selective Piezo modulator.” Not true. Gsmtx4 is an excellent modulator. Which channels have been checked for a kinetic response to Yoda1 to say it is specific for Piezo?

This sentence is not present in the text. We do agree that the effects of Yoda1 on other channels is unknown.

L49 “An avenue to treat these diseases would be to correct Piezo channel activity with small 50 molecule agonists and/or antagonists.” But you just said the channels are everywhere so what could be the consequence of giving something so general?

We did briefly mention that selective modulators may help treat some mechanopathologies related to Piezo channels (L290-292). There are many ways to deliver drug to specific areas of the body to reduce side-effects. For instance, inflammation-induced hyperalgesia has been linked to up-regulation of Piezo2 activity in the skin. Hence, local application of a Piezo2-selective inhibitor, for example using a cream, may help ease pain associated with mechanical hypersensitivity without disrupting normal Piezo2 function in other tissues. However, we are far from having such a drug. Hence, we have shortened this sentence L293-293: “Understanding this interaction will represent a unique opportunity to design selective modulators against Piezo channels.”

L136+ put derivation in supplement

All our derivations were in supplementary files.

How do you know that yoda does not bind to the bilayer and thereby alters stress on the channel?

Fair point. We do not know the contribution of a bilayer effect. However, the fact that Yoda1 does not modulate Piezo2 and the fact that the effect on Piezo1 are totally abolished by small chemical changes of the Yoda1 molecule such as dichloro substitutions and oxidation of the thioether group (Syeda et al. 2015) strongly suggests these effects are mediated by a ligand binding site and not by a lipid effect. We have added a paragraph to discuss this point L312-318.

How are you accounting for the internal kinetics of Ca⁺² handling? That affects kinetic analysis.

We did not account for internal kinetics of calcium handling. We recorded the fluorescence value 30sec after adding the agonist. We found that most cells produce a maximal fluorescence signal within 5-15 sec after agonist application while their fluorescence level remained relatively constant until the end of the 30sec recording period. We occasionally observed very small run-down of calcium signals after the maximal fluorescence has been reached (Fig 1c and Supplementary video 1), which is quite similar to the Yoda1-induced calcium fluorescence signals reported by Syeda et al. (Fig 1 of their paper). This indicates that internal depletion of calcium ions by HEK293T cells within the 30 sec time frame is minimum and also relatively constant across different laboratories. Hence, we do not anticipate internal calcium handling variability in HEK293T cells to significantly affect our measurements.

The title is misleading. Something like “YODA1 interaction with regions of Piezo1” is better

Our current work does not describe the nature of the interaction between the ion channel and the small molecule agonist. Therefore, our title “Probing the Gating Mechanism of the Mechanosensitive Channel Piezo1 with a Small Molecule Agonist” seems appropriate.

My pdf of the paper does not display videos

Reviewer #3 (Remarks to the Author):

I find the paper "Probing the Gating Mechanism of the Mechanosensitive Channel Piezo1 with a Small Molecule Agonist" very informative. The experiments have been carefully done and presented. The paper is written thoughtfully.

By bringing in Yoda1, a small activator molecule, the authors go beyond purely tension-dependent aspects of Piezo1 gating and probe the possible cooperativity and allostery in its mechanism.

General:

The allosteric effect of Yoda1 binding on the open-state energy should include contributions from all subunits, whether they gate cooperatively or not.

We agree with the reviewer that the effect of Yoda1 binding on the free energy of opening should include contributions from all subunits. The fact that the interaction of Yoda1 with only one Yoda1-sensitive subunit is sufficient to stabilize the open state does not imply only one subunit contributes to the total free energy change.

The free energy cost of channel opening cannot be decomposed into three separate subunits. As previously described for mechanosensitive ion channels (BJ 86, 2846), the free energy of opening may be decomposed into a tension-dependent area dilation term $\gamma\Delta A$ and the change in protein energy ΔG_0 :

$$-k_B T \ln \left(\frac{P_{\text{open}}}{P_{\text{close}}} \right) = \Delta G = G_{\text{open}} - G_{\text{close}} = -\gamma\Delta A + \Delta G_0$$

γ is the membrane tension, ΔA is the change in the cross-sectional areas of Piezo1 upon activation and ΔG_0 corresponds to the channel opening free energy.

Based on the law of Laplace $\gamma = pr/2$, in which r is the radius of patch curvature and p is the pressure gradients across the patch, we can speculate that the effect of Yoda1 on the open-state energy may be estimated by $\Delta\Delta G_0 = (pr - p'r')\Delta A/2$, in which p' and r' are the pressure and curvature needed to maintain the same open probability in presence of Yoda1. Further studies will be needed to validate this approach.

We have removed our hypothetical energy diagram (old Fig 8b) because it may be confusing in this regard and because it is mainly supported by data obtained by Syeda et al. The new conclusion Figure 9 illustrates in a simpler fashion the two possible gating mechanisms proposed from our study.

The authors may consider three additional experiments.

(1) Because the energy difference between the open and closed states is directly reflected in P50, it would be good to see the dependence of P50 on Yoda1 concentration, covering the entire range below and above the k_d (~10 μm).

This is a very good suggestion and we have included additional results showing the modulation of P50 by varying Yoda1 concentration in the new figure 7. The plots were fitted with the binding equation (eq 2) and yielded similar apparent affinities between WT and hybrid channels. We have added a new paragraph L271-281 to discuss these new data.

(2) If the authors took careful measurements of P_o versus pressure (tension) near k_d (10 μm), they expect to see the response of a mixed population with different occupancy by the drug. Not

only P50 should be different from the saturating Yoda concentration, but the slope factor k should be lower (see the reasoning in BJ 86 (2004) 2846).

The slope factor of the I/I_{max} vs. pressure plots were obtained by fitting each plot with eq (1) and reported in the Supplementary Table 1. We agree with the reasoning that a more heterogeneous population of active channels should yield “shallower” plots while a more homogeneous population should yield steeper plots. At Yoda1 concentrations near the apparent K_d (3-10 μM), each WT mPZ1 subunit may or not interact with Yoda1, creating a diverse population of ligand-protein complexes. In contrast, at saturating Yoda concentrations, all subunits are necessarily interacting with Yoda, reducing molecular diversity. We do observe a steeper plot with 100 μM Yoda1 ($k \approx 7$ mmHg) but we did not see significant change in the slope factor when reducing concentration near and below the apparent K_d ($k \approx 10$ mmHg). We do not know why. Our best explanation resides with the fact that these cell-attached pressure clamp measurements are obtained in living cells and thus may be affected by cellular factors (e.g. cytoskeleton, membrane microdomains, etc...) that ultimately affect how Piezo channels respond to external pressure.

(3) When the measurement is taken on WT:Chim 1:1 population, the slope factor k even at saturating Yoda concentrations should be measurably lower compared to either WT or Chim uniform populations, for the same reason of non-uniformity. It would be surprising to see if this is not the case.

We do observe a larger slope factor for WT:Chim=1:1 at 100 μM Yoda1 ($k = 13$ mmHg vs. 7 mmHg for WT), consistent with the notion that a more diverse channel population obtained when mixing subunits produces shallower plots (i.e. high k values). However, the slope factor was not too different for lower Yoda1 concentrations between WT and hybrid channels ($k \approx 9-11$ mmHg). As stated above, we do not have a clear explanation as to why the slope factor does not increase further for measurements taken near the apparent K_d . Such slope analysis may be better performed in an in vitro system having less experimental variability (e.g. Piezo reconstituted into giant liposomes).

I have several more particular suggestions that may improve readability of this paper.

Fig. 1. The vertical scale in panels d and e is the same, but the applied stimuli are different. In the first case ch1961 is totally inactive, and fully active in the latter. For the ease of figure readability it would be desirable to put the words ‘osmotic activation’ on panel d.

Thank you, we agree this could be confusing. We have added “osmotic activation” above panels e and h.

Please correct symbols on Fig. 2c.

Thank you for catching this, we have corrected the symbols.

The results presented on panels f and g are not discussed in detail further in the paper, yet they are interesting. They suggest that the binding site immediately coordinating Yoda1 I is likely formed by the C-terminal domain apparently involving Ch1 segment, which binds the drug initially with a higher affinity, but fails to produce full activation. The ‘normal’ activating effect requires a larger chunk of protein, imposing lower affinity at low saturations, highly consistent with the allosteric mechanism.

Thank you for this comment, this interpretation is quite interesting and certainly worth further investigation. Here, we chose not to over-interpret these data for one major reason: since Yoda1 modulates the Piezo1 pressure-sensitivity, the Yoda1-induced calcium-sensitive fluorescence signals obtained in absence of external mechanical stimuli are necessarily dependent on the intrinsic pressure sensitivity of each chimera which have not been determined

expect for “Chim”. We indeed focused on the only chimera exhibiting a total loss of Yoda1-sensitivity for practical reasons. Further investigations of our sub-domain chimera using pressure-clamp electrophysiology will be helpful in identifying the binding mechanism of Yoda1. We have added a comment L110-114 to discuss the importance of the C-terminal part of this region in mediating chemical activation of Piezo1.

It would be desirable to mention the magnitude of osmotic drop in the shock experiments right in the text.

Yes, we have added this sentence L82: “...by reducing extracellular osmolarity by approximately 250mOsmol/L (see Material and Methods).”

Line 123: I would suggest putting: ‘...in the absence of external mechanical stimuli...’ because we don’t know the distribution of membrane tension at rest.

Thank you for correcting this sentence, we have made that change.

Lines 126-132: please mention whether these were on-cell or excised patches and the pipette size.

L129-132, we have modified our sentence to:

“...we performed on-cell pressure-clamp electrophysiology recordings (Figure 2a-c). In these experiments, channel opening is evoked using brief negative pressure pulses applied to the backside of a patch pipette (2-3 μ m tip diameter) while maintaining a patch potential of -80 mV relative to the inside of the cell.”

Line 136, eqn 1: is the midpoint designated as P1/2 or P50? Choose one.

Thank you, we have kept P50 throughout the manuscript to be consistent.

Line 136: ‘ $I = I_{max}/2$ and the slope factor k , a constant expressed in mmHg, that ...’

Thank you, we have modified the sentence L138-139

Line 178: ‘... making it very difficult to link...’ – the authors may also mention the problem that the C- and N- termini may not be close by thus precluding concatenation without a long unnatural linker.

Thank you, this is indeed another fair argument against concatenation. We have added this sentence L193-194 to highlight this point.

Lines 183-185 and all electrophysiology figures: The authors may comment on the difference in channel adaptation times between WT and Chim.

We have compared the inactivation time course between WT and Chim in presence or absence of 30 μ M Yoda1 in the new panels Fig 2e-f. We discuss the difference in inactivation/adaptation in the text L152-158.

Line 209: ‘...WT subunits contribute to the Ca²⁺-induced fluorescence signal...’

Thanks! Corrected L225

Line 285: ‘... some of the residues may form a classical Yoda1 binding site...’

We have changed the sentence L309-311 to: “...hence some of the residues may form a Yoda1 binding site or may allosterically couple the interaction of Yoda1 from a distant binding site.”

Line 322: the

Thanks! Corrected L353

REVIEWERS' COMMENTS:

Reviewer #1 (Remarks to the Author):

The authors have addressed most but not all (see below) of my points; moreover they provide no additional information on the structural mechanism(s) of Yoda1-mediated channel activation. I understand it is beyond the scope of this study, but an Editorial question is whether the 'goal' of identifying 'Yoda1-insensitive mutants' and 'how many agonist-sensitive subunits are needed to mediate agonist activation of Piezo1' is worth publishing in Nat. Com.

One important point still not clear is the lack of detectable

inward current induced by Yoda1, given the (apparently) huge and sustained increase in intracellular calcium seen after Yoda1 exposure. It does make sense to me, given that piezo is more permeable to Na⁺ than Ca²⁺. What is the intracellular free calcium concentration reached in response to Yoda1 (10-100 μM)? Could it be that Yoda1 is activating piezo located in ER? This referee would love to see an example of WCR with 100 μM Yoda1.

Reviewer #3 (Remarks to the Author):

I am satisfied with the revision.

My only request is to add one phrase in the text describing figure 7. The phrase should state that at high Yoda concentration the activation slope is higher (7 mm Hg) than at low concentration (10 mm Hg) apparently due to higher homogeneity of the channel population.

Responses to the reviewers

For clarity, our response are indicated in blue.

Reviewer #1 (Remarks to the Author):

The authors have addressed most but not all (see below) of my points; moreover they provide no additional information on the structural mechanism(s) of Yoda1-mediated channel activation. I understand it is beyond the scope of this study, but an Editorial question is whether the 'goal' of identifying 'Yoda1-insensitive mutants' and 'how many agonist-sensitive subunits are needed to mediate agonist activation of Piezo1' is worth publishing in Nat. Com.

Piezo proteins are a relatively new family of ion channels playing many important biological functions, yet their gating mechanism remains poorly understood. The study presented in this manuscript investigates, for the first time, the subunit contribution to channel opening, therefore providing critical insights into how the trimeric channel operate its permeation pathway. We believe this work to be highly significant.

One important point still not clear is the lack of detectable inward current induced by Yoda1, given the (apparently) huge and sustained increase in intracellular calcium seen after Yoda1 exposure. It does make sense to me, given that piezo is more permeable to Na⁺ than Ca²⁺.

There are at least two reasons that explain why, in presence of Yoda1, no large current has been detected in whole cell recordings in absence of mechanical stimulation (however, a change of steady-state open probability is clearly detectable in single channel recordings, see Syeda et al. 2015).

First, the fluorescence intensity varies as a function of the absolute free intracellular calcium concentration, but the ionic current varies as a function of the rate of change in ionic flow. Hence, a sustained increase of free intracellular calcium does not mean a sustained ionic current. In fact, the peak fluorescence appears at about 10-15 seconds after agonist application, which indicates a very small underlying calcium current.

To explain this point, let's consider a Piezo-expressing cell with a volume of 100 μm³ that increases its free calcium concentration from 100 nM to 100 μM after 15 seconds exposure to Yoda1. The number of free calcium ions in the cell increases from 10³ to 10⁶ in 15 seconds. Thus, 999,000 calcium ions have been flowing in the cell in 15 seconds, which corresponds to the movement of 66,600 calcium ions per second, or 133,200 electrical charges per second assuming a constant current. This charge corresponds to 8.06 x 10⁻¹³ coulomb. This would translate to a whole cell calcium current of 8.06 x 10⁻¹³ A which correspond to a steady-state whole-cell current of only 0.8pA (This value is an underestimation because only the contribution of calcium ions was considered in the calculation). Even if this current was 10-fold larger, it will be very difficult to detect above electrical noise in whole-cell recordings.

A second reason is that the GCaMP6m indicator is exquisitely sensitive to even small changes in free calcium in the sub-micromolar range (see below), which is the typical range of free calcium in a resting cell. Hence very small variations in free calcium concentration can lead to big fluorescence changes.

What is the intracellular free calcium concentration reached in response to Yoda1 (10-100 μM)?

We have not directly measured intracellular free calcium concentrations induced with Yoda1. However, the fluorescence response of GCaMP6m as a function of free calcium concentration has been previously studied by Barnett et al. (PLOS One 2017). In their report, a moderate change in free calcium from 100 nM to 351 nM corresponds to a relatively large $\Delta F/F_0$ of 4, which is similar to the maximal signal we obtain with saturating Yoda1 concentrations. This underscores the very high sensitivity of GCaMP6m to small variations of free calcium (see our response above).

Could it be that Yoda1 is activating piezo located in ER?

The Yoda1-induced calcium-sensitive fluorescence signal is drastically reduced in presence of extracellular EGTA (a calcium chelator) but unaffected by application of the SERCA-inhibitor thapsigargin (Syeda et al. 2015). Hence, Yoda1 does not elicit an increase of cytoplasmic free calcium through Piezo in intracellular organelles but through Piezo located at the plasma membranes.

This referee would love to see an example of WCR with 100 μ M Yoda1.

This experiment was previously attempted by Syeda and colleagues without success at 30 μ M. It is unlikely the result will be different using 100 μ M for reasons explained above. It is however possible to detect an increase of steady-state single channel open probability evoked by Yoda1 in absence of mechanical stimulation using cell-attached patch-clamp experiments (see for instance Syeda et al., eLife 2015).

Reviewer #3 (Remarks to the Author):

I am satisfied with the revision. My only request is to add one phrase in the text describing figure 7. The phrase should state that at high Yoda concentration the activation slope is higher (7 mm Hg) than at low concentration (10 mm Hg) apparently due to higher homogeneity of the channel population.

Thank you, we have added a statement to indicate that the increase slope is consistent with an increase heterogeneity of the channel population and quoted the corresponding reference.